

# An emerging viral pathogen truncates population age structure in a European amphibian and may reduce population viability

Lewis J. Campbell[1,2,3], Trenton W.J. Garner[2], Giulia Tessa[4], Benjamin C. Scheele[5], Amber G.F. Griffiths[6], Lena Wilfert[7,8] and Xavier A. Harrison[2]

[1] Environment and Sustainability Institute, University of Exeter, Penryn, UK
[2] Institute of Zoology, Zoological Society of London, London, UK
[3] Department of Pathobiological Sciences, University of Wisconsin-Madison, Madison, WI, USA
[4] Life Sciences and Systems Biology, University of Turin, Turin, Italy
[5] Fenner School of Environment and Society, Australian National University, Canberra, ACT, Australia
[6] FoAM Kernow, Penryn, UK
[7] Centre for Ecology and Conservation, University of Exeter, Penryn, UK
[8] Institute of Evolutionary Ecology and Conservation Genomics, Universität Ulm, Ulm, Germany

Corresponding authors
Lewis J. Campbell,
lewis.campbell@wisc.edu
Xavier A. Harrison,
Xav.harrison@gmail.com

## ABSTRACT

Infectious diseases can alter the demography of their host populations, reducing their viability even in the absence of mass mortality. Amphibians are the most threatened group of vertebrates globally, and emerging infectious diseases play a large role in their continued population declines. Viruses belonging to the genus *Ranavirus* are responsible for one of the deadliest and most widespread of these diseases. To date, no work has used individual level data to investigate how ranaviruses affect population demographic structure. We used skeletochronology and morphology to evaluate the impact of ranaviruses on the age structure of populations of the European common frog (*Rana temporaria*) in the UK. We compared ecologically similar populations that differed most notably in their historical presence or absence of ranavirosis (the acute syndrome caused by ranavirus infection). Our results suggest that ranavirosis may truncate the age structure of *R. temporaria* populations. One potential explanation for such a shift might be increased adult mortality and subsequent shifts in the life history of younger age classes that increase reproductive output earlier in life. Additionally, we constructed population projection models which indicated that such increased adult mortality could heighten the vulnerability of frog populations to stochastic environmental challenges.

# INTRODUCTION

The emergence of infectious diseases can truncate the age structure of host populations (*Jones et al., 2008*; *Lachish, McCallum & Jones, 2009*; *Ohlberger et al., 2011*; *Fitzpatrick et al., 2014*). Within age-structured populations, such truncation is

primarily caused by compensatory changes in the vital rates (rates of growth, fecundity, and survival) of younger age classes which occur in response to elevated levels of extrinsic adult mortality (death of adult animals attributable to external factors such as disease or predation; *Stearns, 1992*; *Roff, 1993*). To maximise individual fitness within an environment of high extrinsic adult mortality, selection for increased developmental rates, decreased size, and age at sexual maturity, and an increased adult life span (decreased intrinsic adult mortality) will occur (*Stearns et al., 2000*). This theory has been empirically borne out in a number of systems in response to several different sources of mortality, including predation (*Reznick, Bryga & Endler, 1990*), over-harvesting (*Olsen et al., 2004*), and experimentally induced adult mortality (*Stearns et al., 2000*).

Changes to population demography can have profound impacts on the growth and stability of infected populations (*Saether & Bakke, 2000*). This phenomenon has recently been demonstrated by *Scheele et al. (2016)*, who documented high adult mortality and associated truncation of population age structure in alpine tree frog (*Litoria verreauxii alpina*) populations infected with the lethal fungal pathogen *Batrachochytrium dendrobatidis*. Populations with truncated age structure were shown to be more vulnerable to decline due to stochastic recruitment failure than *B. dendrobatidis*-free populations (*Scheele et al., 2016*). These results highlight an important, yet relatively unexplored mechanism by which infectious diseases can negatively affect their hosts.

Infectious diseases are emerging at a faster rate and threatening a larger range of species than at any prior point in history (*Jones, Patel & Levy, 2008*). Therefore, it is imperative to better understand demographic shifts associated with disease, and their consequence for population viability, particularly in species of conservation concern.

Worldwide, amphibians are the most imperilled class of vertebrates (*Wake & Vredenburg, 2008*). Aside from threats such as habitat loss (*Cushman, 2006*), over-harvesting (*Xie et al., 2007*), and climate change (*Foden et al., 2013*), one major driver of amphibian declines is the emergence of a suite of infectious diseases (*Daszak et al., 1999*). One of the most widespread and deadly of these diseases, ranavirosis, is caused by viral pathogens belonging to the genus *Ranavirus* (*Chinchar, 2002*; *Green, Converse & Schrader, 2002*). Ranaviruses are globally distributed and are capable of infecting and killing a wide range of species from three classes of ectothermic vertebrates (Amphibians, *Cunningham et al., 1996*; Fish, *Whittington, Becker & Dennis, 2010*; Reptiles, *Marschang, 2011*). Clinical ranavirosis is often characterised by severe dermal ulcerations (*Cunningham et al., 1996*), as well as haemorrhaging and lesions affecting the internal organs (*Cunningham et al., 1996*; *Bayley, Hill & Feist, 2013*). At the population level, acute incidences of ranavirosis often manifest in mass mortality events, with extreme or complete population mortality reported in some instances (*Green, Converse & Schrader, 2002*; *Wheelwright et al., 2014*). Evidence from the UK shows that following an episode of mass mortality due to ranavirosis European common frog (*Rana temporaria*) populations can decline in size by more than 80% (*Teacher, Cunningham & Garner, 2010*). Though population recovery has been observed in some instances, often population size remains supressed, and some populations decline to local extinction (*Teacher, Cunningham & Garner, 2010*). *Campbell et al. (2018)*

demonstrated that ranaviruses are potentially more widespread than previously thought, and present within the environment even when resident frog populations show no overt signs of disease. These findings suggests that whether or not an outbreak of ranavirosis occurs may depend on currently unknown biotic or abiotic factors of an environment, although there is evidence for a role of the cutaneous or environmental microbiome (*Campbell et al., 2018*), secondary host species, and chemical usage (*North et al., 2015*). Susceptibility to ranavirosis outbreaks varies among species, but a number of ecological risk factors, including life history strategy (the species specific cycle of growth, maturation, reproduction and death), have been identified (*Hoverman et al., 2011*). To date, no work has evaluated the impact of ranavirosis on the demographic structure of host populations using individual age data.

In this study, we used a unique comparative field system born out of the Frog Mortality Project (FMP; see *Teacher, Cunningham & Garner, 2010*; *Price et al., 2016* for details) to study the impacts of ranaviral disease history on the demographic structure of wild *R. temporaria* populations in the UK. Unlike in most other species of susceptible amphibian, ranavirosis disproportionately kills adult *R. temporaria*, rather than tadpoles (*Duffus, Nichols & Garner, 2013*). We determined population age structure using skeletochronology (age determination by counting skeletal growth rings) to test the hypothesis that a history of ranavirosis truncates the age structure of *R. temporaria* populations. Additionally, we investigated potential mechanisms for such age truncation by using morphometric data to explore if frogs originating from populations with a history of ranavirosis display reduced body size or evidence of more rapid growth compared to their counterparts from ostensibly disease-free populations. Finally, to examine the potential effect of the demographic impacts of ranavirosis, we used population projection modelling to simulate the dynamics of hypothetical *R. temporaria* populations under a range of stochastic environmental scenarios that impact recruitment, adult survival, or both. Based on previous evidence (*Ohlberger et al., 2011*; *Rouyer et al., 2012*; *Scheele et al., 2016*), we hypothesised that age structure truncation would heighten the vulnerability of *R. temporaria* populations to environmental stochasticity.

## METHODS

### Ethics statement

This project was approved by the ethics boards of both the University of Exeter and Zoological Society of London and conducted under UK Home Office project license 80/2466. All field sampling was conducted under the personal Home Office license 30/10730 issued to Lewis Campbell.

### Field sampling

Evidence from the same populations studied herein suggests that ranaviruses may be more ubiquitous within amphibian populations in the UK than previously thought, being detectable even in populations with no history of ranavirosis (*Campbell et al., 2018*). We therefore used a history of ranavirosis related mortality (or lack of), rather than

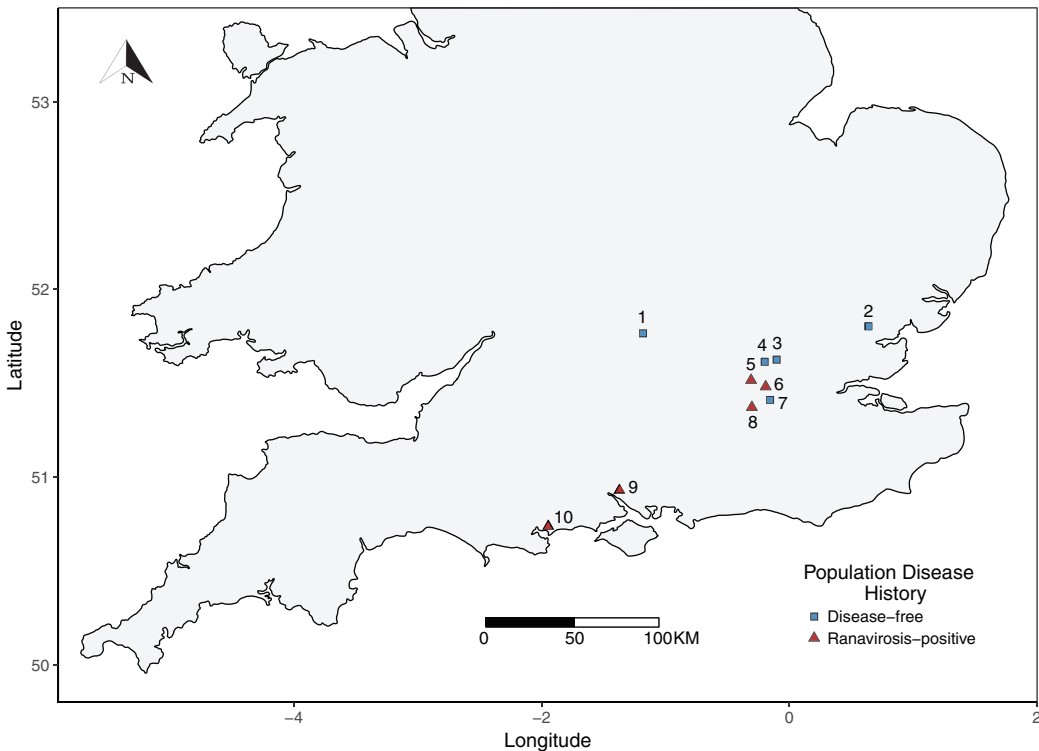

**Figure 1 Map of sampled *R. temporaria* populations.** Map of the locations of sampled populations within the southern UK. Fieldsites were drawn from the Frog Mortality Project database of populations known to have experienced mass mortality events due to ranavirosis and a complimentary database of populations known to have been ranavirosis-free since disease emergence in the 1990s. Populations = 1. Oxford; 2. Witham; 3. Palmer's Green; 4. Folkington Corner; 5. Ealing; 6. Chessington; 7. Mitcham; 8. Tadworth; 9. Southampton; 10. Poole.

detectable ranavirus burden, to differentiate between populations that are impacted by ranaviruses and those which are not. Candidate *R. temporaria* populations were identified using the FMP database of privately owned fieldsites that have experienced at least one *R. temporaria* mass mortality event due to ranavirosis and a complimentary database of fieldssites thought to have not experienced a ranavirosis outbreak, based upon information provided by fieldsite owners. All fieldsites were urban or sub-urban gardens, distributed throughout Southern England (Fig. 1). Although sizes varied, all fieldsites contained an artificially constructed pond. Site owners reported that neither their gardens nor ponds were treated with any chemicals which could harm the resident *R. temporaria* populations.

Briefly, ranavirosis-positive fieldsites were classified as such if they had experienced a confirmed ranavirosis mass mortality event between 1997 and 1998 and ongoing discovery of frog carcasses with symptoms consistent with ranavirosis was reported by property owners to *Teacher, Cunningham & Garner (2010)*. Ranaviruses were confirmed as the causative agents of mortality within these populations using a combination of post-mortem pathological examination of frog carcasses and molecular diagnostic techniques (*Teacher, Cunningham & Garner, 2010*). In order to be classified as disease-free, fieldsites must have been owned or occupied by the same people at all times since 1997 and have

contained a well-monitored *R. temporaria* population for at least as long. Owners/occupiers of disease-free fieldsites must not have observed any signs or symptoms of ranavirosis at any time. See *Teacher, Cunningham & Garner (2010)* for more detailed fieldsite selection criteria. Fieldsite proprietors were contacted to establish their willingness to be involved in this study. Five ranavirosis populations were successfully recruited and matched with five populations that have remained disease-free (Fig. 1). None of our ranavirosis-positive populations had experienced a mass mortality event within the last decade.

Fieldsites were surveyed during the spring breeding season of 2015, with each site sampled on a single day. Sampling involved opportunistic searching for frogs within and around the perimeter of the breeding ponds. Captured frogs were placed into plastic holding tanks before sampling, which took place *in situ*. To ensure sampling effort between populations was as equal as possible, searching took place during a 1 h, mid-morning time window.

Snout to vent length (SVL) was measured using 0.1 mm scale callipers and the distal portion of the 1st digit of a hind limb was clipped using surgical scissors. A topical disinfectant that contained an analgesic (Bactine; WellSpring Pharmaceutical, Sarasota, FL, USA) was applied to the surgical area prior to the procedure. Toe clips were placed into individual 1.5 ml micro-centrifuge tubes containing 1 ml of 70% ethanol. Following sampling, all animals were released at point of capture. The number of individuals sampled in each population varied between 4 (Witham) and 61 (Mitcham and Palmer's Green) with a mean of 30 animals sampled per site (Table S1). All captured frogs were considered to be part of the breeding population, as juvenile frogs rarely return to breeding ponds (*Wilbur, 1980*; *Verrell, 1985*), and all sampled individuals were found to be over the minimum known age of sexual maturity for the study species (*Gibbons, 1983*; *Miaud, Guyétant & Elmberg, 1999*).

## Age determination

The age of each frog was determined by skeletochronology, which has been calibrated and demonstrated as a reliable method of determining the age of northern European *R. temporaria* in several previous studies (*Gibbons, 1983*; *Gibbons & McCarthy, 1986*; *Ryser, 1996*; *Miaud, Guyétant & Elmberg, 1999*). We followed the protocol for aging *R. temporaria* from (*Miaud, Guyétant & Elmberg, 1999*) with the following modifications. The phalangeal bone was separated from soft tissues, decalcified with 5% nitric acid for 1.5 h, and washed with water over night. Cross sections (12 μm thick) were then cut from the bone using a cryostat and stained using haematoxylin for 20 min. Lines of arrested growth (LAG) were counted using a light microscope at 200–400× magnification, 10–12 sections were analysed for each individual and two different researchers verified counts. Age at sexual maturity was determined as the youngest age at which inter-LAG space reduced in size, as juvenile inter-LAG space is significantly wider than post-sexual maturity (*Sinsch, 2015*).

## Statistical analyses

### Body size by age and age at sexual maturity

We conducted all statistical modelling in R (*R Core Team, 2014*). We used linear mixed effects regression (lmer) models, implemented in the package lme4 (*Bates et al., 2015*),

**Table 1 Summary of statistical model simplification procedure and results.**

| Fixed effects structure | Removed fixed effect | Est | df | Chi$^2$ | p |
|---|---|---|---|---|---|
| Male body size | | | | | |
| svl~age * status | | | 6 | | |
| svl~age + status | Age * status | | 5 | 0.059 | 0.81 |
| svl~age | Status | | 4 | 0.21 | 0.65 |
| svl~1 | Age | 3.51 | 3 | 186.38 | <0.001 |
| Female body size | | | | | |
| svl~age * status | | | 6 | | |
| svl~age + status | Age * status | | 5 | 0.94 | 0.33 |
| svl~age | Status | | 4 | 2.43 | 0.12 |
| svl~1 | Age | 3.21 | 3 | 47.16 | <0.001 |
| Male age at maturity | | | | | |
| agemat ~ status | | | 4 | | |
| agemat ~ 1 | Status | −0.12 | 3 | 0.99 | 0.32 |
| Female age at maturity | | | | | |
| agemat ~ status | | | 4 | | |
| agemat ~ 1 | Status | −0.12 | 3 | 0.29 | 0.59 |

Notes:
  Model summaries of model simplification procedure to evaluate the effect of ranavirosis history on the body size and age at maturity of *R. temporaria* populations. All models contained only population of origin as a random effect applied to model intercepts.
  The *p*-values presented here represent the significance of the parameter removed from the preceding model as calculated by a likelihood ratio test between models (ANOVA in R).
  svl, snout to vent length; agemat, age at sexual maturity; df, degrees of freedom of the model; Est, effect size estimates of the final fixed effect removed from each model. All comparisons use disease-free populations as the reference level.

fitted with a Gaussian error structure, and a stepwise simplification procedure to investigate the impact of population ranavirosis history on the body size (SVL) of *R. temporaria*. Age, ranaviral disease history of the source population (a two level binary factor) and their interaction were fitted as fixed effects. We controlled for variation in SVL between sampled populations by the inclusion of population of origin as a random effect, applied to the intercepts (Table 1). Since male and female frogs grow at different rates (*Gibbons, 1983*; *Ryser, 1996*; *Miaud, Guyétant & Elmberg, 1999*), the datasets of each sex were analysed separately.

A separate lmer model was fitted to investigate the impact of ranavirosis history status of the source population on age at sexual maturity. In the full model, age at maturity was fitted as the response variable, ranaviral disease history of source population as a fixed effect and source population as a random effect. As female *R. temporaria* mature later than males (*Gibbons, 1983*, *Miaud, Guyétant & Elmberg, 1999*), the datasets of each sex were analysed separately.

### Influence of ranavirus on population age structure

The impact of disease status on the age structure of *R. temporaria* populations was investigated using a Bayesian ordinal mixed effects model in the package MCMCglmm (*Hadfield, 2010*). We fitted age class as an ordinal response variable (nine discrete classes, ages 2–10 years), disease status of the source population as the fixed effect, and source
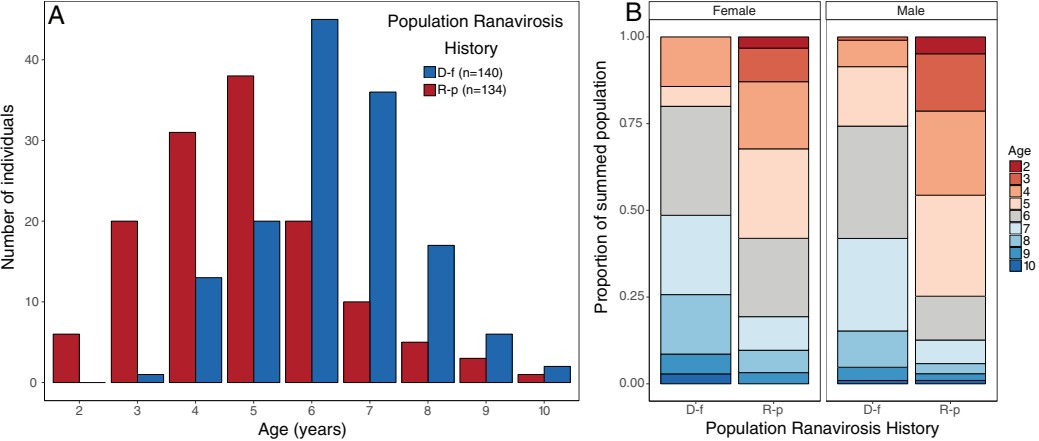

**Figure 2 Observed age structure across disease-free and ranavirosis-positive *R. temporaria* populations.** (A) Histogram of raw counts of numbers of individuals observed per age class per disease history status type. (B) Proportional stacked bar chart of the proportion of individuals found in populations of each disease history that was a given age, broken down by sex. Breeding populations with a positive history of ranavirosis are dominated by animals 5 years of age and younger. Disease-free populations are majorly comprised of animals 6 years of age and older. D-f = Disease free, R-p = Ranavirosis-positive.

population as a random effect. We used uninformative priors for both the random effect (G) and residual variance (R) structures, but fixed the residual variance at 1 as this quantity cannot be estimated in ordinal models (*Hadfield, 2010, 2018*). The model was run for a total of 600,000 iterations with a burn-in period of 100,000 iterations and a thinning rate of 500, giving a final sample of 1,000 draws from the posterior distributions. We assessed model convergence using the Gelman–Rubin (G–R) statistic calculated from three independent chains initiated with overdispersed starting values. All G–R values were <1.05, indicating convergence. Mean probability of membership and associated 95% credible intervals for each age class were calculated from the linear predictor, for each of the two disease history groups. Age structure plots (Fig. 2) suggest that observed changes were similar for both sexes, so the dataset was not split by sex.

## Population matrix modelling

To investigate how changes in population age structure, as well as scenarios that can bring about such changes, can impact the dynamics and stability of *R. temporaria* populations, we constructed population matrix models. Comprehensive methodologies of our matrix modelling can be found in our supplementary methods section (Methods S1). However, in brief, we created hypothetical *R. temporaria* populations of 150 sexually mature female animals and projected these populations 20 years into the future based on two matrices which represented potential vital rates at ranavirosis-positive *R. temporaria* population (increasing annual mortality in each adult age class) and a disease-free *R. temporaria* population (uniform adult mortality). We populated these matrices using the vital rates for *R. temporaria* published by *Biek et al. (2002)* except that we increased the fecundity of each post sexual maturity age class of *R. temporaria* by 50 eggs compared to the previous age class. Theoretical clutch sizes ranged from 250 eggs per year

per 2-year-old adults to 650 eggs per year per 10-year-old adults. This increase was done according to age by size data collected for this study and previously published work showing that the clutch sizes of *R. temporaria* are tightly positively correlated with female body size with a >3 fold increase between individuals of 50 mm versus individuals of 80 mm (*Gibbons & McCarthy, 1986*). To analyse the impact of these adjustments we compared the sensitives of our adjusted matrices to one constructed with the unadjusted vital rates of *Biek et al. (2002)*. As previous evidence has shown that disease induced demographic shifts or the changes in vital rates which may bring them about can increase susceptibility to environmental change (*Rouyer et al., 2012*; *Scheele et al., 2017*) we then modelled our populations in environmentally stochastic scenarios which incorporated processes which may result in annual reproductive failure, mass mortality, or in extreme cases, both simultaneously.

## RESULTS

### Body size by age and age at sexual maturity

We sampled 208 male and 66 female frogs, of which 103 males and 31 females were sampled at ranavirosis-positive populations. A break-down of the number of frogs sampled per each population can be found in Table S1. For both sexes, age had a significant effect on SVL (males; d$f$ = 4, Chi$^2$ = 186.38, $p$ < 0.001, females; d$f$ = 4, Chi$^2$ = 47.16, $p$ < 0.001; Fig. S1; Table 1). Mean age at sexual maturity of males from ranavirosis-positive populations ($n$ = 57) was 2.6 years (± SE 0.07) and from disease-free populations ($n$ = 59) it was 2.8 years (± SE 0.06). Mean age at sexual maturity of females from ranavirosis-positive populations ($n$ = 19) was 3.2 years (± SE 0.13) and for females from disease-free populations ($n$ = 17) it was 3.3 years (± SE 0.10), and all differences were non-significant (Fig. S2; Table 1).

### Influence of ranavirosis on population age structure

The mean age of males from ranavirosis-positive populations was 4.8-years-old (± SE 0.16) compared to an average age of 6.3 years (± SE 0.13) at disease-free populations. Mean female age at ranavirosis-positive populations was 5.3 years (± SE 0.28) compared to 6.6 years (± SE 0.26) at disease-free populations.

Disease history had a significant effect on population age structure (effect size −1.43, 95% credible intervals −2.37, −0.38, $p$ = 0.008; Fig. 2). We found populations with a positive history of ranavirosis to be dominated by younger *R. temporaria*. We calculated the difference in posterior probabilities of belonging to an age class based upon disease history status by subtracting the posterior probability of a frog of age $X$ being encountered at ranavirosis-positive population from the posterior probability of a frog of age $X$ being found at a disease-free population. The resulting difference values show that adults aged 2–5 years old are more likely to be encountered in positive disease history populations, and those aged 6–10 years old are more likely to be observed at populations where no disease has been recorded (Fig. 3). Differences in age distributions were strongly supported for all age classes (95% credible intervals of difference do not cross zero) except for 6-year-olds. Although 6-year-old frogs were more likely to be

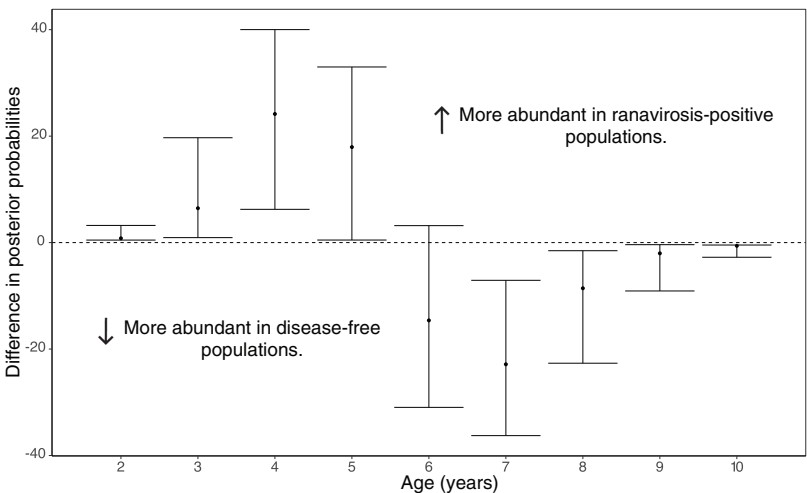

**Figure 3 Differences in posterior probabilities of belonging to a given age class between population groups of varying disease history.** The mean difference in the posterior probabilities of belonging to a given age class by population ranavirosis history. Values >0 indicate that an age class is more likely to be observed in a ranavirosis-positive population and <0 a disease-free population. An age with 95% (2.5–97.5%) credible intervals that do not span zero suggests that influence of disease history on that age class is significantly supported by our model. This is the case for all classes other than age 6 which although found to be observed more often in disease-free populations has credible intervals spanning 0.

observed in disease-free populations, the 95% credible intervals incorporated 0 (mean difference = −14.41, 95% CI [−30.45 – 3.96]; Fig. 3).

## Population projection modelling

When projected based upon a matrix with constant annual adult survival, both disease-free and ranavirosis-positive population vectors showed similar population dynamics. Population growth initially spiked due to an influx of pre-mature age classes into the population. This initial growth was followed by a period of attenuating oscillation, before reaching equilibrium at around the 15th year (Fig. S3). Projected populations starting with a truncated age structure obtained smaller population sizes per each year, however, this difference was non-significant (Fig. S3, ANOVA; d$f$ = 1, $F$ = 0.935, $p$ = 0.34). Projecting either starting population vector using a matrix incorporating reduced annual adult survival resulted in the same pattern of population growth but further reduced population sizes per year, with the lowest annual population sizes obtained by an age truncated population projected with decreased annual adult survival, though this reduction was again non-significant (Fig. S3; ANOVA; d$f$ = 3, $F$ = 0.99, $p$ = 0.39).

Elasticity analysis showed that both disease-free and ranavirosis-positive matrices were similarly sensitive to the same matrix elements. However, the decreased annual adult survival incorporated into our ranavirosis-positive matrix rendered that matrix marginally more sensitive to fluctuations in survival of larval, juvenile and 2-year-old life stages. Sensitivity to survival of all subsequent adult age classes was higher in our disease-free population matrix (Fig. S4). Similarly, our ranavirosis-positive matrix was more sensitive to changes in the fecundity of 2, 3 and 4 year old breeding adults, whereas sensitivity

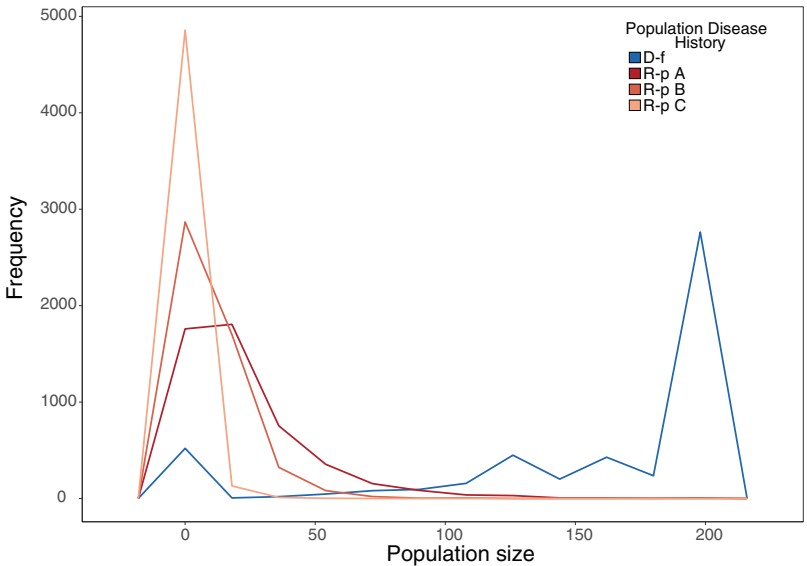

**Figure 4 Frequency plot of the number of model iterations in which each modelled population reached a given population size under stochastic environmental conditions.** Frequency polygon of iterations in which the projected population hit a given size in stochastic projection modelling. The same starting population vector based on summed observed disease-free populations was used in all models. D-f = Simulated disease-free population under a 10% annual chance of complete reproductive failure. R-p A = Simulated ranavirosis-positive population under a 10% annual chance of complete reproductive failure. R-p B = Simulated ranavirosis-positive population under a 10% annual chance of reproductive failure AND a 10% annual chance of a recurrent adult mass mortality event in exclusive years. R-p C = Simulated ranavirosis-positive population under identical conditions to R-p B = with addition of a 5% annual chance of complete recruitment failure and adult mass mortality in the same year.

to changes in fecundity of older adult age classes was higher in our disease-free population matrix (Fig. S5). In concordance with the matrix constructed using the unadjusted vital rates of Biek et al. (2002), both of our altered matrices were most sensitive to fluctuations in the summed vital rates of post metamorphic *R. temporaria* life stages, though larval survival demonstrated the highest or second highest individual elasticity in all matrices (Fig. S4). The most noticeable dissimilarity between our adjusted matrices and the unadjusted matrix was that matrix sensitivity was distributed more evenly between the fecundity of all adult age classes in the former, whereas the unadjusted matrix was extremely sensitive to the fecundity of 2-year-old breeding adults (Fig. S5).

Our stochastic projection models showed that disease-free populations subject to a 10% probability of recruitment failure per year were still able to consistently attain population sizes near carrying capacity much more often than ranavirosis-positive populations subjected to the same scenario, which attained a more variable array of population sizes (Fig. 4; Table 2). The inclusion of potential ranavirosis mass mortality events further destabilised ranavirosis-positive populations and under a scenario when both adult mass mortality and reproductive failure could occur within the same year populations were driven locally extinct in 58% of model iterations (Fig. 4; Table 2).

**Table 2 Summary of the number of projection model iterations in which a population reached a given size under stochastic environmental conditions.**

| Model | Extinct | <50 | 50–100 | 100–150 | 150–199 | K |
|---|---|---|---|---|---|---|
| Disease-free | 12 | 525 | 177 | 734 | 1,319 | 2,233 |
| Ranavirosis-positive A | 429 | 4,062 | 443 | 48 | 15 | 3 |
| Ranavirosis-postiive B | 611 | 4,300 | 87 | 2 | 0 | 0 |
| Ranavirosis-positive C | 2,918 | 2,081 | 1 | 0 | 0 | 0 |

**Note:**
The number of iterations per 5,000 that each stochastic projection model reached a given population size. Disease-Free = Simulated disease-free population under a 10% annual chance of complete reproductive failure. Ranavirosis-positive A = Simulated ranavirosis-positive population under a 10% annual chance of complete reproductive failure. Ranavirosis-postiive B = Simulated ranavirosis population under a 10% annual chance of reproductive failure and a 10% annual chance of a recurrent adult mass mortality event in exclusive years. Ranavirosis-positive C = Simulated ranavirosis-positive population under identical conditions to Ranavirosis-positive B with addition of a 5% annual chance of complete recruitment failure and adult mass mortality in the same year. $K$ = Imposed population carrying capacity of 200.

## DISCUSSION

### The impact of disease on population demographics

Our results illustrate that a history of ranavirosis can have a significant effect on the age structure of a population. We found that older *R. temporaria* (aged 6–10 years) are significantly less likely to be found within populations with a positive history of ranavirosis than they are in disease-free populations. Since none of our sampled populations have experienced a significant mass mortality event within the lifetime of any of the sampled frogs (~10 years), this suggests a pattern of attritional mortality in ranavirosis-positive populations, rather than the sudden and catastrophic loss of any particular age classes. Attritional mortality of adults is consistent with our knowledge of ranavirosis in *R. temporaria*. First, *R. temporaria* are highly philopatric (*Brabec et al., 2009*), and our study populations occupy permanent, urban, or semi-urban garden ponds (*Teacher, Cunningham & Garner, 2010*; *Price et al., 2016*). Ranaviruses have been shown to persist in such water bodies (*Nazir, Spengler & Marschang, 2012*), particularly in the presence of secondary host species such as fish or other, less susceptible, amphibian species (*Hoverman et al., 2011*; *North et al., 2015*). None of our fieldsites supported natural or stocked populations of fish, however, we did encounter the common toad (*Bufo bufo*) at one population of each disease status. The presence or absence of secondary hosts therefore does not represent a consistent confounding variable between our disease history groups. Second, mortality due to ranavirosis is annually recurrent (*Daszak et al., 1999*; *Teacher, Cunningham & Garner, 2010*), and unlike all other host-ranavirus systems, infection in the UK primarily affects adult life stages of *R. temporaria* (*Cunningham et al., 1996*; *Duffus, Nichols & Garner, 2013*). Third, adaptive immune response to ranaviruses are apparently limited in *R. temporaria* (*Price et al., 2015*; *Campbell et al., 2018*). Consequently, it is possible that adult mortality within populations with a history of ranavirosis is maintained in such a way that an individual is more likely to become infected and succumb to ranavirosis the more often it returns to spawn. Such a scenario would explain the decreased likelihood of observing individuals older than 5 years of age in populations with a history of ranavirosis.

Concurrently, we found that a significantly higher number of 2–5 year old frogs were captured at disease-free populations than at populations with a history of ranavirosis.

Life history theory predicts that the first compensatory response to high adult mortality should come in the survival rates of lower age classes (*Stearns, 1992*). We lack any data on the immature age classes present at our study populations and the snapshot nature of our study means we cannot draw inference on whether the survival rates of younger adult age classes are increased in ranavirosis-positive populations. However, it is plausible that such an increase could result in the observed increased abundance of younger animals.

Empirical studies conducted on populations of other vertebrates subjected to persistent infectious diseases have detected a reduction in the age or size at sexual maturity within diseased populations (*Jones et al., 2008*; *Lachish, McCallum & Jones, 2009*; *Ohlberger et al., 2011*; *Fitzpatrick et al., 2014*). As such, we hypothesised that *R. temporaria* originating from populations where ranavirosis causes increased adult mortality would reach sexual maturity at an earlier age than frogs from disease-free populations. We also hypothesised that a trade-off in the allocation of resources to early reproduction, away from growth, would cause frogs from ranavirosis-positive populations to attain a lower body size per age than those from disease-free populations. However, we found that a positive history of ranavirosis is not associated with a significant impact on either age at sexual maturity or body size throughout life. These findings are in contrast with the findings of (*Scheele et al., 2017*) who found that an infectious disease (*B. dendrobatidis*) reduced the size and age at sexual maturity of *L. v. alpina.*

The age at which *R. temporaria* mature is intrinsically linked to attaining a minimum body length needed to successfully reproduce (*Ryser, 1996*; *Miaud, Guyétant & Elmberg, 1999*). The age at which this length is attained has been shown to be heavily influenced by several environmental factors such as photo-period and altitude (*Miaud, Guyétant & Elmberg, 1999*). All frogs in our study were found to mature at either 2, 3, or 4 years of age and this is consistent with the findings of other similar studies on *R. temporaria* (*Gibbons, 1983*; *Ryser, 1996*; *Miaud, Guyétant & Elmberg, 1999*). In fact, no previous study has found male or female *R. temporaria* to reach maturity younger than 2 years of age and post-sexual maturity there is little detectable trade-off between growth and fecundity in response to sub-prime environments (*Lardner & Loman, 2003*). This evidence suggests that the life history strategy of *R. temporaria* may already be optimised to generate maximum reproductive fitness in light of other environmental factors and that scope for further plasticity in traits such as age at sexual maturity and subsequent growth rate in response to diseases may be minimal.

Given the apparent lack of compensatory change in the onset of sexual maturity an alternative explanation for the fact that we encountered 2- and 3-year-old breeding frogs at ranavirosis-positive populations but not disease-free populations may be behavioural changes that enhance the life time reproductive success of individuals in the face of high adult mortality. Participation in spawning events in exposed aquatic environments is associated with a significant mortality risk to an individual, caused either by exposure to predation or the act of mating itself (*Beebee, 1996*). Additionally, smaller *R. temporaria* present in breeding populations are easily outcompeted by larger individuals, who are better able to secure a mate and achieve amplexus (*Gibbons, 1983*). The loss of larger, more

competitive individuals at ranavirosis-positive fieldsites may release smaller *R. temporaria* from this intraspecific competitive pressure. Increased opportunity for less competitive individuals to participate successfully in reproductive events may result in animals that would normally defer breeding until subsequent years or larger body sizes attempting to reproduce earlier. Additionally, in environments of high adult mortality the number of lifetime reproductive events is potentially limited. In such environments lifetime reproductive success is likely higher when an individual exploits all possible chances to produce offspring. Thus smaller, less competitive individuals may attempt to breed earlier than they would do in the absence of disease induced mortality, irrespective of the odds of being out competed by older/larger frogs.

Any or all of the potential processes outlined above could lead to the observed truncation of age structures in *R. temporaria* populations with a history of ranavirosis. While our results provide strong evidence of such truncation, they are based on data collected from adult *R. temporaria* sampled during one breeding season. As such we are unable to draw conclusions as to the mechanisms which drive observed changes to population age structure. Additionally, due to the correlational nature of our study we are unable to emphatically exclude the possibility that the age structure truncation we observed is attributable to some other unmeasured variable between our studied populations. Although the lack of systematic differences between our fieldsites other than disease history suggests that this is unlikely. Nevertheless, our results highlight a potentially important and as yet unexplored facet of the relationship between ranaviruses and their amphibian hosts. The impact of demographic shifts has been demonstrated in a number of study systems (*Jones et al., 2008*; *Lachish, McCallum & Jones, 2009*; *Ohlberger et al., 2011*; *Rouyer et al., 2012*; *Scheele et al., 2016*) and as such the relationship between ranaviruses and population demographics undoubtedly warrants further investigation. A long-term, mark-recapture study within the same populations used here would likely prove a critical first step in elucidating the mechanisms which bring about shifts in *R. temporaria* population age structure.

## The potential impact of disease on the viability of populations

Previous investigations have shown that age structure truncation, associated with an emerging pathogen, severely reduces the viability of host populations, particularly under variable environmental conditions (*Scheele et al., 2016*). We used population matrix models to probe the potential impacts of a truncated age structure, and the demographic processes which may bring about such changes, on the viability of *R. temporaria* populations in the UK. Additionally, we introduced environmental stochasticity into our models in order to explore the combined impact of these two challenges simultaneously.

When modelled in the absence of increased adult mortality due to ranavirosis, starting population vectors representing both a disease-free population and a ranavirosis-positive population (truncated age structure) exhibited minimal difference in their dynamics. This finding suggests that the level of age structure truncation we document is unlikely to reduce the viability of *R. temporaria* populations. The same was
true when both population age structures were modelled in the presence of increasing adult mortality due to ranavirosis, suggesting that even under such conditions the relatively high survival of pre-metamorphic, juvenile and young adult age classes is enough to maintain population viability. This suggestion is supported by the elasticity of our population matrices which demonstrated that the survival of tadpoles, juveniles and early adult life stages have higher predicted impact on the population growth dynamics of our populations than the survival of older adult age classes. This result is supportive of the findings of other investigations of amphibian population dynamics (*Biek et al., 2002*; *Earl & Gray, 2014*).

However, under environmentally stochastic scenarios, where recruitment was reduced, populations that were subjected to increasing adult mortality due to ranavirosis faired much worse than ranavirosis-free populations. These results are consistent with previous observations of age structure truncation increasing the vulnerability of populations to environmental stochasticity (*Ohlberger et al., 2011*; *Rouyer et al., 2012*; *Scheele et al., 2016*). Body size, age and fecundity are positively correlated in *R. temporaria* (*Gibbons & McCarthy, 1986*), as well as in many other species (*Blueweiss et al., 1978*; *Honěk, 1993*; *Trippel, 1993*; *Sand, 1996*; *Penteriani, Balbontin & Ferrer, 2003*). The relative absence of the oldest and largest breeding animals from disease positive populations means that *per capita* fecundity will be reduced and annual recruitment rates lowered. Such changes likely heighten the impacts of any events that result in failed recruitment or further adult mortality.

It is important to note that our matrix models are approximations of the populations that they represent, and the results of our models are therefore demonstrative of potential additional impacts of ranavirosis, rather than definitively demonstrating the existence of such impacts. Although our simulations are based on published literature, they necessarily incorporate several assumptions. A key assumption in our models is the number of eggs produced per each individual, per each adult year of age. Although we have strong grounds to suggest that fecundity of female *R. temporaria* increases with age and size, the number of eggs produced by each frog of a given age or size is unknown in our study populations. If a larger number of eggs are produced than represented by our matrices, it may be that the impact of stochastic recruitment failure upon our populations is reduced and vice versa. Similarly, we reduce the survival of each adult age class into the next by 5% for each year of development post-sexual maturity. Whilst we found no impact of the amount by which we reduced this annual survival parameter on the dynamics of our modelled populations, in reality survival may not decrease annually in such a uniform manner. Finally, we set an upper limit for our theoretical populations of 150 female animals. However, due to the nature of our fieldsites we lack the ability to make accurate estimations of actual population sizes at our studied populations. Larger actual population sizes may buffer populations against the potential impacts that we have demonstrated but smaller actual population sizes may mean that the potential demographic impacts of ranavirosis may be even more consequential for *R. temporaria* populations. Further investigation of the life history traits and an accurate estimation of sizes of our study populations would allow for more robust parameterisation of our

models and the ability to more accurately predict the impact of ongoing ranavirosis within them.

Despite these limitations, our population models corroborate previous empirical data collected from within our study system. A long-term study of population sizes has shown that following an outbreak of ranavirosis, characterised by a mass mortality event, UK *R. temporaria* populations follow three possible trajectories. These are: (1) complete recovery to post outbreak population levels, (2) persistence at a largely reduced population size, or (3) local extinction (*Teacher, Cunningham & Garner, 2010*). Our simulated *R. temporaria* populations were projected to population sizes that incorporate all of these possible outcomes, dependant on the levels of environmental stochasticity to which a population was subjected. Importantly, our simulations suggest that the fate of a population subjected to ranavirosis may depend heavily upon the stability of its environment, providing a potential explanation as to why some populations appear to persist with endemic infectious disease, while others are driven to local extirpation.

### The importance of variations in fecundity in population models

Variations in adult fecundity due to body size are often not included in population projection modelling (*Briggs et al., 2005*; but see *Zambrano et al., 2007*). However, our results highlight the importance of considering age or body size specific changes in fecundity in population modelling, particularly when considered threats disproportionately impact certain age classes. Ensuring matrix models of populations represent the life history of the study species as closely as possible is essential, especially when seeking to inform conservation or policy decision making.

### CONCLUSION

Our results highlight an increasing need to better understand the impact of disease on the demography of host populations and the processes which can bring about such demographic shifts. Further investigation of this relationship, possibly via a long-term mark-recapture study on the same populations used here could help elucidate the exact mechanisms responsible for age structure truncation in *R. temporaria* populations. This work also further suggests that the emergence of an infectious disease within a population can heighten its vulnerability to external stressors. Although the theoretical stressor incorporated into our models was environmental stochasticity, the same is likely to be true for all types of stressor including anthropogenic. This result is timely given that we live in a time of unprecedented disease emergence and anthropogenic change (*Daszak, Cunningham & Hyatt, 2001*).

### Funding

This work was supported by a Natural Environment Research Council PhD Studentship. The funders had no role in study design, data collection and analysis, decision to publish, or preparation of the manuscript.

## Grant Disclosures

The following grant information was disclosed by the authors:
A Natural Environment Research Council PhD Studentship.

## Competing Interests

Xavier A Harrison serves as an Academic Editor for PeerJ. Amber G F Griffiths is a director of FoAM Kernow (FoAM Kernow is an affiliated studio of FoAM, which is a multinational, not for profit, network of transdisciplinary labs at the intersection of art, science, nature and everyday life).

## Author Contributions

- Lewis J. Campbell conceived and designed the experiments, performed the experiments, analyzed the data, contributed reagents/materials/analysis tools, prepared figures and/or tables, authored or reviewed drafts of the paper, approved the final draft.
- Trenton W.J. Garner conceived and designed the experiments, contributed reagents/materials/analysis tools, authored or reviewed drafts of the paper, approved the final draft.
- Giulia Tessa performed the experiments, contributed reagents/materials/analysis tools, authored or reviewed drafts of the paper, approved the final draft.
- Benjamin C. Scheele analyzed the data, authored or reviewed drafts of the paper, approved the final draft.
- Amber G.F. Griffiths authored or reviewed drafts of the paper, approved the final draft.
- Lena Wilfert authored or reviewed drafts of the paper, approved the final draft.
- Xavier A. Harrison analyzed the data, prepared figures and/or tables, authored or reviewed drafts of the paper, approved the final draft.

## Animal Ethics

The following information was supplied relating to ethical approvals (i.e., approving body and any reference numbers):

This work was approved by the ethics committees of both the University of Exeter and the Institute of Zoology (Home Office Project License: 80/2466; Home Office Personal License: 30/10730).

## Field Study Permissions

The following information was supplied relating to field study approvals (i.e., approving body and any reference numbers):

Permits were granted by the UK Home Office (Home Office Personal License: 30/10730).

## Data Availability

GitHub: https://github.com/zoolew/Ranavirus-FrogDemography

## Supplemental Information

Supplemental information for this article can be found online at http://dx.doi.org/10.7717/peerj.5949#supplemental-information.

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
