# Peer review of "An emerging viral pathogen truncates population age structure in a European amphibian and may reduce population viability"

_PeerJ, doi:10.7717/peerj.5949_

## Round 0.1 · original submission · Minor Revisions

I would ask you to please provide a point-by-point indication of how your revised manuscript responds to each of the reviewers' detailed comments. I would also like to to pay particular attention to reviewer 1's concerns related to the validity of the findings.

·

Basic reporting

I found the manuscript to be written clearly and concisely, the authors provide sufficient citation of previous work, and the figures and tables are easy to read understand.

The hypotheses are clearly stated and justified with prior work with Bd.

Experimental design

Generally, I found this study to employ a unique data collection strategy (measuring age of adult frogs using bone fragments and counting lines of arrested growth), which leads to an interesting study. The statistical analyses and modeling were also clearly presented and justified. All code is provided for review.

I do have some minor questions/comments about the statistical analyses, which can be seen in the provided code, but should be made clear in the main text as well:

1. Line 134: I believe "Sex" should be changed to "Age"

2. Is the history of ranaviral disease a simple binary? (0/1)

3. Was the random effect of source population applied to the intercept, the slopes, or both?

4. Related to the above, I believe table 1 should include estimates of the slope coefficients for the best models, so that the readers can understand trends in the positive or negative direction. Alternatively, these could be provided in the supplement.

5. Line 147: The residual variation was fixed to 1. Is this a convention in the literature? If so, could a citation be provided? If not, how sensitive are the results to such an assumption?

6. Why were only 2 chains used in the MCMC? I strongly encourage at least 3 chains to assess convergence and the lack of divergent likelihood profiles in the sampler.


For the population matrix model simulations, (line 232), could the authors comment on how often it might occur that there is an adult die-off event and there is zero recruitment in the same year? Has this ever been documented for this frog species? It seems like a very extreme event.

Validity of the findings

I do have some concerns with the conclusions of this manuscript, as currently presented in the text.

1. My main concern is that the age structure, which clearly differs between the ranavirosis-positive and negative sites, cannot be directly linked to ranaviral disease, as this is a correlative study. However, no alternative hypotheses or explanations are provided, and not enough information is given about these field sites to adequately assess how clear the link is between ranavirus die-offs, which occurred at least 10 years ago, and current age-structure.

Specifically, the authors allude to a study that shows that ranaviruses are likely present in many sites, but this does not always lead to disease. In other words, ranavirosis has some determinants other than the presence of the virus. Because of this, it leads me to wonder: (1) were ranaviruses ever found at any of the ranavirosis-negative sites? and (2) if so, what differs between these sites that might influence disease and that might, therefore, influence the age-structure of the frogs, independent of the disease?

Due to this issue, I believe the authors should provide sufficiently more detail about their field sites and what biotic and abiotic aspects might differ between sites that could explain the age-structure difference, either in combination with or independently of historical ranavirosis.

In addition, in line 9 of the abstract, the authors state that the populations "differed only in their historical presence or absence of ranavirosis". This exemplifies my point, as, surely, differences exist between sites other than ranavirosis history.

2. My second concern is with the population viability simulations. Lines 165-168 suggest that the authors somehow changed the data from Biek et al. (2002) to substantially increase the egg contribution from very old adults (10 years old). First, I think the authors could clarify this sentence, as I was confused as to what exactly the authors mean. Did the authors assume some sort of linear relationship between age and egg count? How exactly is this different from Biek's work? I bring this up because of the authors' results that environmental stochasticity and disease combine to greatly affect population viability. I'm wondering how sensitive these results are to the authors' assumption of very large egg counts of the very old individuals that die due to virus? I do recognize that the authors do a good job tempering their simulation results (starting line 347).

Reviewer 2 ·

Basic reporting

Overall the article does wonderful job of meeting criteria under "Basic reporting". I've noted a few specific instances where this could be slightly improved:

Line 16: remove the extra “of” in the middle of this line.
In line 22, please rephrase your meaning of “selection for increased juvenile survival” or add a bit of explanation/ context. This is a bit confusing as selection for traits can result in increased juvenile survival, but survival is typically not a trait that can be selected for. To further this point, study objectives stated in lines 72-78 do not suggest that selection FOR juvenile survival was examined, but rather that things affecting juvenile survival (body size, growth rate) were examined.
Line 50: I think there are some more recent studies that demonstrate ~ 100% mortality at the population level. I believe the following literature may be useful:
• Wheelwright NT, Gray MJ, Hill RD, Miller DL (2014) Sudden mass die-off of a large population of wood frog (Lithobates sylvaticus) tadpoles in Maine, USA, likely due to ranavirus. Herpetol Rev 45:240–242
• Petranka JW, Murray SM, Apple Valley MN, Kennedy CA (2003) Response of amphibians to restoration of a southern Appalachian wetland: perturbations confound post-restoration assessment.
• Todd-Thompson M (2010) Seasonality, variation in species prevalence, and localized disease for ranavirus in Cades Cove (Great Smoky Mountains National Park) amphibians. Master Thesis, University of Tennessee, Knoxville. http://trace.tennessee.edu/utk_gradthes/665 .
Line 52-53: This statement needs a citation.
Line 95-96: I think there is a grammatical error or a word omitted. Perhaps it should read, “…consistent with ranavirosis SUGGESTED continued disease.”
Line 127: missing “and” after the comma.
Line 160-161: It looks like part of the line of text is missing (the “S” is left off of “Supplementary methods”). Perhaps there was an error during conversion from a Word processing document to a PDF.

Experimental design

Solid experimental design.
The population projection modeling seems like a wonderful way to potentially identify areas or scenarios where populations may be at a greater risk of extinction associated with Rv.
The authors do a great job of explaining the limits of their work (e.g. line 351-355). A couple minor things to address:
(1) Lines 135-137: Looking at Table 1, I believe the modeled interaction is not Sex, as stated in line 134, but rather Age. I think this is likely just a type-o.
(2) I think this article may benefit from a more detailed description of the assumptions the authors make in regards to the disease status of a pool. An important aspect is to discuss if any lab verification (pathology, PCR, other diagnostic techniques) or other methods used to determine that the causal linkage between Ranavirus and symptoms. I believe other amphibian diseases could (perhaps in concert) cause symptoms that are similar if not indistinguishable from Ranavirosis. Providing any additional information about this linkage (or lack thereof) would be useful to clarify what it means for a pool to be “Ranavirosis-positive” – even if it’s just to say that causation is assumed and no testing leading to verification of Rv infection occurred for these study pools. It would be similarly useful to note if any past testing had or had not detected in Rv in the “disease free” pools actually had Rv present? (I.e., detectable yet not causing observable disease)
Additionally, even though no mass mortality events were detected in “Ranavirosis-positive” pools within the last decade, it may be helpful to describe the continued pathogen and disease presence by providing some summary statistics about incidence of disease even if it didn’t result in a mass mortality event. E.g., Frequency of years that diseased individuals been detected? How long since the last detected diseased individual? # of diseased individuals detected? Etc.

Validity of the findings

One quick comment:
Line 246: they reference persistence of Rv in the presence of secondary host species. Were there any secondary host species identified in the study pools? Knowing this would make this comment in the discussion more relevant.

Additional comments

I believe this paper is a great contribution to the literature relevant to understanding how Rv impacts amphibian communities. I really enjoyed reading this as it was well-written and presented timely information.

---

## Round 0.2 · accepted · Accept

Thank you for your thorough responses to each of the two reviewers' comments, which I believe makes your paper much stronger.

#